# Characterization of Particle Size Distributions and Water-Soluble Ions in Particulate Matter Measured at a Broiler Farm



**Philip J. Silva** [1,*], **Tanner Cress** [2], **Ryan Drover** [2], **Cara Michael** [2], **Gregory Docekal** [3], **Pierce Larkin** [3], **Antonio Godoy** [2], **Devin A. Cavero** [2], **Crystal Sin** [2], **Janise Waites** [2], **Rezaul Mahmood** [3,†], **Martin Cohron** [4] and **Kathleen L. Purvis-Roberts** [2]

1   USDA-ARS, Food Animal Environmental Systems Research Unit, Bowling Green, KY 42104, USA
2   W.M. Keck Science Department, Claremont McKenna, Pitzer, and Scripps Colleges, Claremont, CA 91711, USA; tjcress@illinois.edu (T.C.); rdrov002@ucr.edu (R.D.); caramich@bu.edu (C.M.); agodoy@students.pitzer.edu (A.G.); devincavero@berkeley.edu (D.A.C.); crysin@students.pitzer.edu (C.S.); jwaites22@students.claremont.edu (J.W.); kpurvis@kecksci.claremont.edu (K.L.P.-R.)
3   Department of Earth, Environmental, and Atmospheric Sciences, Western Kentucky University, Bowling Green, KY 42101, USA; gdocekal@woodstockga.gov (G.D.); pierce.larkin@noaa.gov (P.L.); rmahmood2@unl.edu (R.M.)
4   Heritage Farms, 4210 Sugar Grove Rd., Bowling Green, KY 42101, USA; martin.cohron@wku.edu
*   Correspondence: phil.silva@usda.gov
†   Current address: School of Natural Resources, University of Nebraska-Lincoln, Lincoln, NE 68583, USA.

**Abstract:** The chemical composition and size distribution of particulate matter produced at broiler poultry houses is not well understood, so this is a novel study to understand the particulate size distributions at a poultry house as well as the ionic composition of the particulate matter using real-time methods. Two optical particle counters provided particle size distributions inside and outside the house. An ambient ion monitor and a particle-in-liquid sampler analyzed the ionic chemical composition of the particulate matter in the house while a scanning mobility particle sizer provided size information in the nanoparticle range. Ammonia concentrations in the house were measured using a chemical sensor. Ammonia concentrations in the house were consistently in the lower part of the per million range 2–20 ppm. The optical particle counter and ion chromatography measurements both showed a strong diurnal variation of particulate matter concentration in the house throughout the study, associated with the lights being on and animal activity. Particulate mass concentration inside the house was dominated by coarse mode particles as opposed to the outdoor sampler which showed much smaller sizes. A few new particle formation and growth events were observed in the house. Ionic constituents detected by chromatography made up a small fraction of the overall mass concentration. The composition of the ionic constituents was similar for most of the study with typical ions being ammonium, sodium, potassium, chloride, sulfate, nitrate, nitrite, phosphate, and several carboxylates (formate, acetate, propionate, and butyrate.) At the end of the study, bromide was also detected during the last several days. Overall, we determined that the ionic components of the particulate matter formed through secondary particle formation was small, but also that some ionic constituents can be associated with management practices.

**Keywords:** poultry; ammonia; fine particles; coarse particles; secondary aerosol; ion chromatography

## 1. Introduction

Air emissions from concentrated animal feeding operations (CAFOs), are reported to cause health impacts for both the animals and humans exposed [1–3]. The concentration of fine particulate matter (PM$_{2.5}$) and gases emitted from CAFOs can have regional impacts on people living in the area around them [3–7]. More recent studies show that emissions from agricultural facilities can cause the nitrogen enrichment of soils and waterways [8]. While general surveys have been carried out to understand overall gas and PM$_{2.5}$ concentrations

and some chemical speciation of particles on CAFOs, a greater amount of analysis is needed to understand the chemistry for the secondary formation of particulate matter (PM) and PM composition.

Emission factors of CAFOs are small as an overall gaseous pollutant and $PM_{2.5}$ source, but they can be a significant contributor to air quality concerns where CAFOs are concentrated in regional areas. Understanding the concentrations of gases from CAFOs and the chemistry of particle formation can provide additional data to help regional-scale atmospheric chemistry models be more accurate [9]. Traditionally, agricultural sources of emissions have not been included in air quality regulations because of a lack of data and the difficulty of mitigation. In 2021, a court in Maryland became the first to order a state environmental department to regulate air emissions from poultry farms [10]. While this ruling is being contested in appeals, the potential that air emissions from CAFOs may face future regulations emphasizes the importance of having more solid and extensive data.

Different types of poultry farms have been studied, including egg-laying and broiler houses [7,10–14]. Layer houses, both caged and caged-free, showed high concentrations of $PM_{2.5}$, $PM_{10}$, ammonia, and hydrogen sulfide air pollution [13–16]. The chemical composition of $PM_{2.5}$ is different inside poultry houses versus outside, showing that pollutants that are ventilated from the poultry houses can impact particle formation outside [17,18]. The concentration and composition of $PM_{2.5}$ can also change depending on the location of the facility and within the enclosure itself [19,20]. The major sources of fine particulate matter derive from feathers and manure, although feed and bedding materials also contribute to particle mass [20,21]. Not as much research has been carried out to understand emissions from meat chicken sheds, as opposed to egg-laying houses, but odor emissions are thought to be dependent on chicken litter conditions, such as moisture and microbial content of the litter [22]. Litter with higher moisture content appears to lead to higher microbial communities that produce more odorous emissions [23–25].

Secondary particle formation occurs from agricultural sources, but most research to date has focused on large-scale processes such as emissions from fertilizer application and the burning of fields [26–28]. New particle growth occurs from secondary particle formation inside poultry houses, but this formation is not often observed on the short time scales they occur. Gases have been measured and characterized in agricultural settings [15,22,29,30]. Ammonia gas, which is present in high concentrations in CAFOs, forms secondary particles when reacting with sulfate and nitrate in the air, but it is unclear which other gaseous compounds could also form particles. The water-soluble ions studied were standard common cations and anions as well as those chosen based on gaseous compounds identified previously in agricultural settings [29]. Modeling studies have shown the importance of amines on particle formation from agricultural sources [31]. This study probes the difference between secondary particle formation versus the increase in primary particle concentration inside the broiler house.

## 2. Materials and Methods

Air quality collection and analysis were conducted at a poultry farm from 15–30 June 2018 with two houses, 500 ft × 40 ft (152.4 m × 13.7 m), of 25,000 chickens each in the Sugar Grove community of western Kentucky (Figure 1). The poultry houses were ventilated by 16 large metal fans each and were climate controlled. The ambient temperature peaked during the day at temperatures ranging between 29 and 36 °C and fans turned on above 22 °C to keep the birds cool. Conventional wood chip bedding was not changed during the sampling time. The particulate matter measurements were taken when chickens were 5–7 weeks into their growth cycle, and it takes approximately 47 days for chickens to grow to market size. The flock was in the poultry houses from 14 May through 30 June 2018. We would have liked to carry out an air quality study during the life of the flock, but were limited by funding for the field study. The lights were turned off at 10 pm at night and turned on again at 4 am in the morning and the chickens had continuous access to food and

water with adequate ventilation. The chickens were removed from the house for delivery to the processing factory on 30 June.

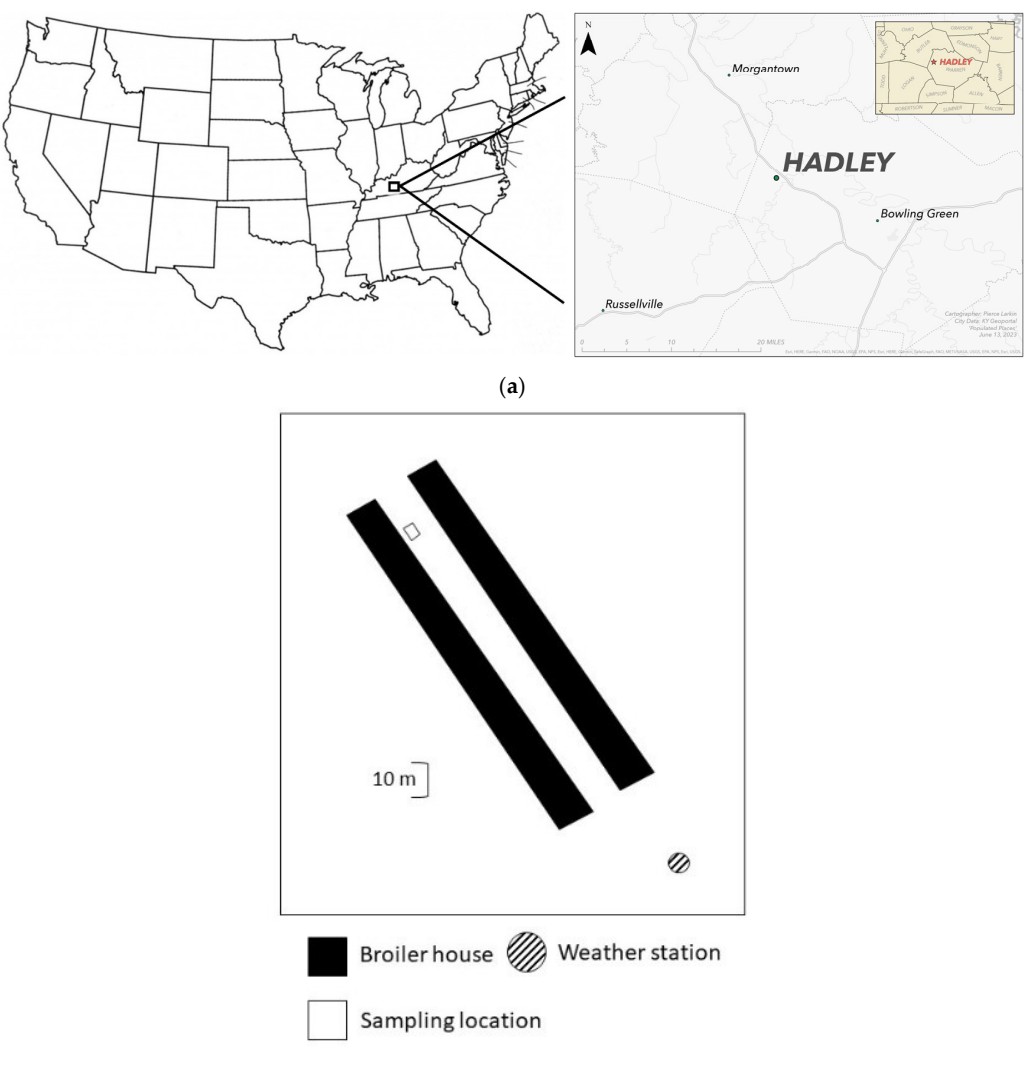

(**a**)

(**b**)

**Figure 1.** (**a**) Location of the Sugar Grove community in relation to Bowling Green, Kentucky, and the United States; (**b**) Layout of broiler houses at the farm.

The goal of the project was to understand whether particulate matter inside and outside the poultry house was created through primary or secondary particulate matter formation processes and to determine the contribution of water-soluble ions to particle formation. The gas phase and particulate measurements were all continuous to understand real-time atmospheric processes. Gas phase sensors were used to measure the concentration of ammonia and hydrogen sulfide as precursor gases to particle formation. Particle size was measured with a scanning mobility particle sizer and concentrations of particles in certain size bins were measured with an optical particle counter. Finally, the concentration of the water-soluble ions was measured with a particle-into-liquid sampler coupled to ion chromatographs and an ambient ion monitor. Details about these instruments are explained below.

Chemical sensors (Cairpol) for ammonia (nitrogen) and hydrogen sulfide (sulfur) monitored the indoor air at the poultry house. While the sensors are typically battery-run and recharged using USB cables, the sensors were kept plugged in to allow for continuous running throughout the experiment. The ammonia sensor has a lower detection limit of 100 ppb while the hydrogen sulfide sensor has a lower detection limit of 10 ppb. The sensors

yielded a running average reading every minute. Gas-phase sulfur measurements will not be discussed here as they were almost always below the detection limit of the sensor.

Particulate size was measured using several instruments. A scanning mobility particle sizer (SMPS) was used for particulate sizes less than ~0.5 μm. The SMPS was a TSI model number 3034 butanol-based SMPS. The SMPS acquired particulate size distribution from 10 nm–0.5 μm, with one scan taken every five minutes. This scan separates particles into 80-size bins over that size regime. The optical particle counter (OPC) is a Met One model number 212 profiler that detects particles via light scattering. The OPC collected data from ~0.3–10 μm, and separated the data into eight size bins. The OPC sampled every 20 s and recorded averaged data once per minute. Two OPCs sampled during the campaign, one was placed inside the house next to the sampling line for the other instruments located in the trailer and the second OPC was located outside the house next to the trailer and about 3 m from one of the exhaust fans of the house.

A particle-into-liquid sampler (PILS) was used for the conversion of particulate matter into liquid samples to be analyzed with ion chromatography [32,33]. The PILS includes a particle growth device and an impactor plate upon which particles are collected. A solution of LiBr was washed across the impaction plate to collect the water-soluble ions and pumped into the ion chromatograph sample loop.

Ion chromatography analysis for the PILS was performed using two Metrohm 761 compact ion chromatography (IC) instruments, one to analyze the cation and the other to analyze anion species (Metrohm AG, Herisau, Switzerland). Ions calibrated for this study were acetate, chloride, formate, nitrate, nitrite, sulfate, magnesium, methylaminium, and potassium. Some of these ions are standard particulate components (e.g., potassium, nitrate) and others were chosen to understand gas-to-particle conversion based on previously identified gases measured in agricultural field studies (amines and carboxylic acids) [29,34,35]. Additional information such as standard procedures, precision and accuracy, minimum detection limits, etc., can be found in previously published papers [33,34,36]. Both ICs were equipped with an injection valve and a low-pulsation dual-piston pump. Both IC units were coupled to a degassing assembly (Phenomenex Degassex Model DG-4400 4-channel online degasser). The column pathway for both cation and anion analysis contained a Metrosep RP guard column (Metrohm AG product number 6.1011.030). The anion chromatograph was equipped with suppressed detection, using a 100 mM $H_2SO_4$ solution. Samples of 500 μL were injected onto the columns. Analysis of cation species was performed through a Metrosep C4 250 mm (Metrohm AG product number 6.1050.430), and the eluent for cation IC analysis was a solution of 3.0 mM nitric acid and 3.5% acetonitrile in Millipore water with a flow rate of 1.00 mL/min. For the anion analysis, a Metrosep A Supp 5 250 mm column was used (Metrohm AG product number 6.1006.530) was used. The eluent for anion analysis consisted of 3.2 mM sodium carbonate and 1.0 mM sodium bicarbonate at a flow rate of 1.00 mL min$^{-1}$. The laboratory reactions were performed at isocratic conditions at 30 min intervals and an exterior temperature of approximately 29 °C.

A Teflon-coated cyclone (URG part number 2000-30EH) was used to select $PM_{2.5}$ at a flow rate of 16.7 L min$^{-1}$. The cyclone was attached to the end of black conductive silicon tubing (TSI, 0.44 In. ID × 0.75 In. OD part number 3001835) inside of the poultry house at approximately 1.8 m height. Annular denuders were used to react away the acidic and basic gases in the inlet line. Each denuder consisted of three annuli with a 1 mm separation of etched quartz glass surfaces of 15.1 cm in length (URG part number URG-2000-30 × 242-3CSS). A 0.15 M sodium carbonate coating solution was used to capture acidic gases and a 0.20 M citric acid coating solution for the basic gases. We found that the citric acid denuder solution is not adequate enough to remove all of the ammonia gas near a high-concentration source, such as an agricultural facility. As such, ammonium concentrations from the PILS-IC analysis will not be discussed.

An ambient ion monitor (AIM) was also used for ion chromatography analysis. The AIM was a URG model D dual ion chromatography-based system that analyzes gas and particle ions. The AIM instrument uses a parallel plate denuder to strip gas phase species

from the sample to convert to ions followed by a steam distillation apparatus to collect water-soluble particulate matter, also for ionic analysis. During this experiment, the AIM was sampled at a standard 3 L min$^{-1}$ and the gas-phase denuder used a standard aqueous solution of 5 mM solution hydrogen peroxide ($H_2O_2$) for the collection of gases. The $H_2O_2$ has been used by multiple users for the AIM in the standard configuration to allow for a more complete conversion of sulfur gases to sulfate ions. Of note, through our own laboratory experiments, we have found that the normal denuder solution is not adequate to allow for the full collection of ammonia when near a strong source such as an agricultural facility, so ammonium concentrations from the AIM are not reliable in this study as there was a carryover from the gas sample into the particulate. The AIM was collected hourly, with 5 mL sample volumes from the denuder and the steam collector for gas and particulate samples, respectively. Samples were injected through ion concentrators for subsequent injection onto ion chromatographs to allow for the higher sensitivity from analysis of the full 5 mL sample. Due to the analysis of a full 5 mL sample per run, the AIM tends to have better sensitivity than the PILs instrument, but can potentially become oversaturated as well from gas and particulate samples of high concentration.

Ion chromatography (IC) analysis on the AIM was accomplished using two ICs; one anion and one cation based. The anion-based chromatograph is equipped with a Dionex AS-11HC column for the analysis of common anions and organic acids. Ions calibrated for this study were fluoride, chloride, bromide, nitrite, nitrate, sulfate, phosphate, formate, acetate, propionate, and butyrate. The cation-based chromatography is equipped with a Dionex CS-19 column for analysis of ammonia and amines as well as common ions (sodium, potassium, calcium, magnesium). Both separations took 28 min allowing for two separations per hour sample period (1 gas sample, 1 particulate sample). Similar to the PILs, the ions detected were a mix of common ions found in particulate matter as well as ions that could be expected from the agricultural environment (amines and carboxylic acids).

A high-volume pump (~30 L min$^{-1}$) and large-diameter manifold (~30 cm) were used for the continuous flow of air from the house to the trailer. The manifold travelled through the house wall for a distance of ~5 m. Individual instrument lines were connected to the manifold using the appropriate line types. The AIM instrument inlet line was a standard Teflon-coated aluminum line, suitable for the detection of both particulate and gas-phase species. The SMPS line utilized black conductive silicon tubing, similar to the PILS-ICs used to minimize electrostatic effects on-small particle-size samples.

Data were collected to monitor ambient conditions outside the broiler operation. A weather station was placed to the south-east, approximately 150 feet from the broiler houses. Temperature, relative humidity, wind direction, wind speed, wind gust speed, rain, and solar radiation measurements were taken every five minutes throughout the study with a weather station starter kit (HOBO U30-NRC) that included a weather station data logger (U30-NRC-SYS-C), silicon pyranometer sensor (S-LIB-M003), light sensor level (M-LLA), and 0.2 mm rain gauge (S-RGB-M002).

It should be noted that this study should not be regarded as a comprehensive analysis of the chemical composition of particulate matter at an animal feeding operation. Only a fraction of the chemical composition of particulate matter at any CAFO will be water-soluble ions that the PILS-IC and AIM can detect. The rest of the particulate mass would include inorganic mineral components, non-water-soluble organic components, and biological components, none of which can be detected by ion chromatography.

## 3. Results

### 3.1. Ammonia

Since neither of the ion chromatography instruments could effectively remove and detect the ammonia present near the source using standard denuders, the chemical sensors, which are designed for high-concentration environments, were set up at the poultry house to measure ammonia and hydrogen sulfide gases. The hydrogen sulfide sensor was consistently near or below the detection limit of 10 ppb and will not be discussed. The

ammonia sensor data have a gap in the middle of the study due to a failure on the data-logging device, but showed concentrations above 1 ppm at most times during the study (Figure 2). The ammonia concentration reached as high as 12 ppm on 18 June. While these are high concentrations compared to the ambient atmosphere, they are quite low concentrations for inside animal housing. On the last day of the study, the NH$_3$ sensor surged and saturated during the removal of the birds from the house.

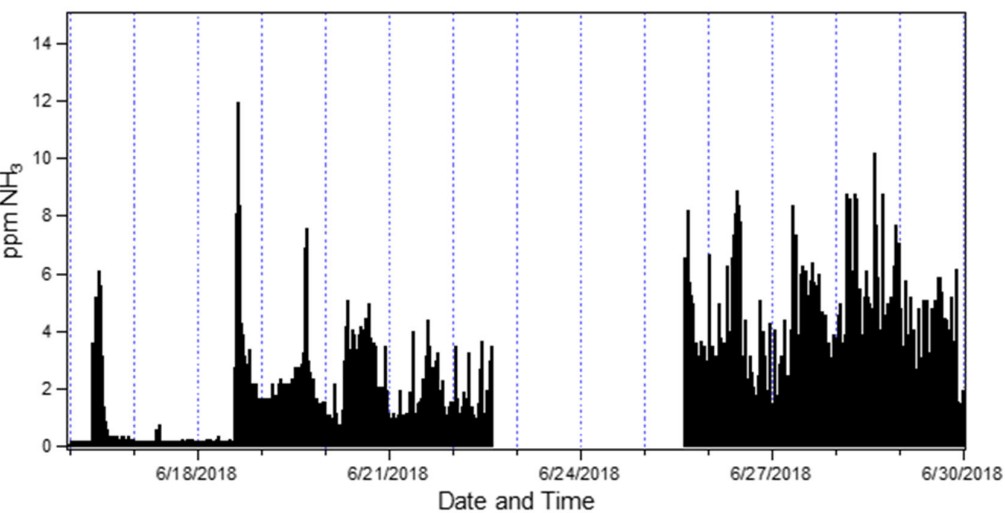

**Figure 2.** Concentration of ammonia over the sampling period.

### 3.2. Particle Sizing

Particulate matter (PM) from agriculture and animal feeding operations are largely considered from the point of view of coarse mode dust particles (PMc), but agricultural emissions consist of a range of reactive gases including ammonia, sulfides, and other volatile organic compounds that could react to form new particles or accumulate on existing particles. This indicates that there may be a potential for a secondary formation of PM. For this study, an SMPS and OPC both sampled particles in the poultry house to provide full-size distribution measurements from 10 nm–20 mm. The SMPS yielded a size distribution every 3 min. In addition, two OPCs were used to sample both inside and outside the house at the main exhaust fan. The OPCs were set to give 1 min data averages in order to record rapid dust events at the poultry house.

### 3.2.1. Optical Particle Counter

OPC data were acquired from outside the house starting from 20 June to the end of the study. A separate OPC was set up inside the house starting on 21 June. Figure 3 shows the mass concentration traces for both OPC instruments using PM$_1$ and PM$_{10}$, assuming unit density spheres for the particulate matter. Gray bars on the plot signify when housing lights were turned off and on, at 10 p.m. and 4 a.m. daily.

Several observations are made from these OPC data. In the indoor data, a strong diurnal trend appears where particle concentrations are significantly higher during the daytime when housing lights are on than during the nighttime hours. This is not a surprise as animal activity during the day results in dust particulate entrainment in the air and is probably the main source of particulate matter in the house whereas the limited activity of animals at nighttime results in less particulate being generated in the air.

The outdoor pattern of particulate matter also has a diurnal pattern. The outdoor data more closely match ambient daylight hours than reflecting the lighting hours indoors. PM$_{10}$ values are always significantly higher indoors than outdoors. Yet such is not always the case at the lowest size range for PM$_1$. At times, the outdoor concentrations are higher than indoors, again reflecting that atmospheric photochemistry outdoors is contributing to

particulate matter formation. Not all of the ambient particulate matter outside is from the agricultural housing, even right next to the source.

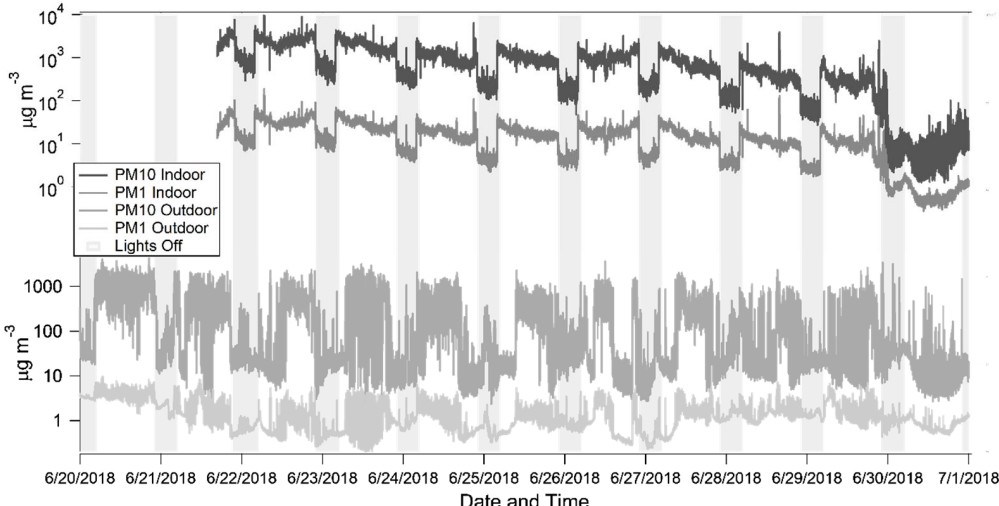

**Figure 3.** Mass concentrations of particulate matter for $PM_1$ and $PM_{10}$ indoors and outdoors at the poultry house using optical particle counters and assuming unit density. Lights were turned off at 10 p.m. and on at 4 a.m. inside the poultry house, represented by the gray bars.

The vast majority of the particulate mass concentration is from the coarse mode particulate matter, which is to be expected, and is largely correlated with animal activity entraining dust particles in the air. However, the OPCs cannot detect the smaller aerosol sizes that would indicate potential particle growth events in the house. For that, the SMPS data are needed. It should be kept in mind that any particle growth events shown on the SMPS, while potentially important in number concentration, are a tiny fraction of the overall particulate mass.

Table 1 shows the fraction of particulate concentration measured by the indoor and outdoor OPCs for $PM_{10}$, $PM_5$, $PM_3$, $PM_1$, and $PM_{0.5}$. The table shows a larger fraction of smaller particles in the outdoor sampling environment. Indoors, only half of the $PM_{10}$ is $PM_5$ and only 3% of the mass is below $PM_1$. In comparison, for the outdoor particulate matter as measured by the OPC, roughly 80% of the $PM_{10}$ is $PM_5$ and ~15% of the $PM_{10}$ is below 1 μm. Some of this is due to the quick half-life of larger particulates settling to surfaces, but this also likely shows that even right next to a house, a significant fraction of PM is from outdoor sources (regional background photochemistry), and not all measured particulate matter is primary particulate emissions directly from the poultry house.

**Table 1.** Fraction of particulate concentrations measured by the indoor and outdoor OPCs.

|  | Indoor Fraction | Outdoor Fraction |
|---|---|---|
| $PM_{10}$ | 1 | 1 |
| $PM_5$ | 0.57 | 0.82 |
| $PM_3$ | 0.27 | 0.57 |
| $PM_1$ | 0.03 | 0.16 |
| $PM_{0.5}$ | 0.003 | 0.03 |

### 3.2.2. Scanning Mobility Particle Sizer

As stated earlier, an SMPS was used to obtain particle size distributions inside the house at the small size ranges (~10–500 nm). The small particle sizes are a tiny fraction of the overall mass concentration (0.3% according to the indoor OPC), but a large part of the number concentration. Overall SMPS data for the sampling period (22–29 June) are shown

below in Figure 4. While these figures can be complex to understand, the *x*-axis shows sampling time, the *y*-axis shows particle size distribution in 80 bins from 10 nm–0.5 μm, and the *z*-axis (color) shows the particulate number concentrations. The data show several particulate growth events where the size of particulates grows slowly (e.g., the days of 24, 25, and 28 June) over the sampling period. The growth can be seen as the distribution shifts slowly upward, representing particulate growth in the green, yellow, and red colors. In addition, several time periods resemble potential new particle formation events.

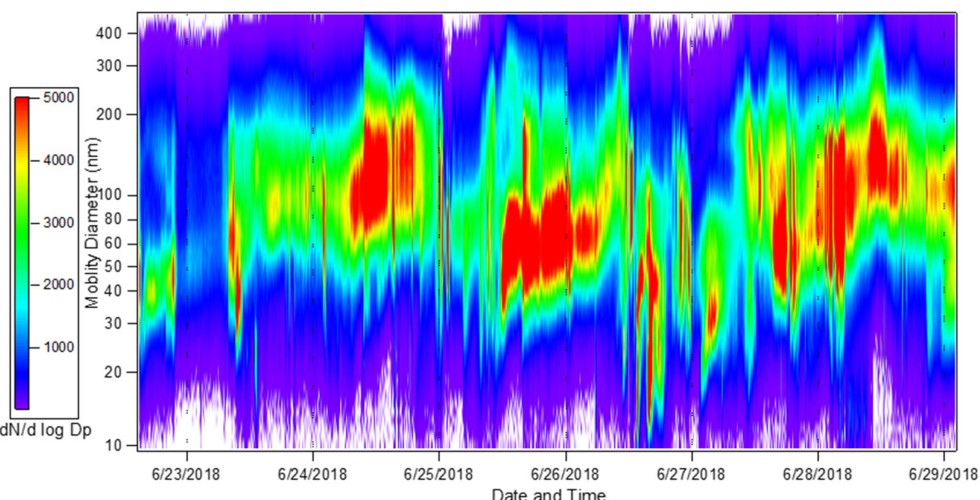

**Figure 4.** Particle number concentration and size distributions determined by SMPS for the study from 22–29 June 2018.

A particle accumulation and growth event (shown in detail in the appendix, Figure A1) was observed 25–26 June. During this period the median particle diameter increased from ~40 nm to ~60 nm over a 12 h period and then increased further to ~120 nm over the next 8 h. Growth periods like this were observed several times during the sampling period.

In addition, several times during the sampling period, new particle formation (npf) events looked like they were observed. One of these was observed on 26 June. A closer view of the npf event on 26 June is shown in Figure A2. We were not expecting any such events due to the very high particulate mass concentrations (and related high particle surface area) to allow for gas-phase accumulation on the already existing particles. However, gas-phase concentrations must have temporarily reached high enough levels to cause new particles to form during this time, in spite of the already high particulate concentrations.

*3.3. Particle Chemical Composition*

In Figure 5, the total concentration of water-soluble PM$_{2.5}$ detected by the PILS-IC and AIM is shown. The PILS-IC and AIM data agree strongly with OPC data that large increases in PM are observed when lights are on inside the house and the animals are active. Gray boxes in the plot denote when lights are off inside the house and data show daily increases in ion concentration during the middle of light hours with minimal concentrations overnight when lights are off. Note that similar to the sizing data, most of the mass concentration is due to animal activity during the day resulting in the production of aerosols.

Although the PILS-IC and AIM instruments both measure aerosol ions using ion chromatography, the methods are sufficiently different so that direct comparison of ions is not possible in many cases. Because the AIM instrument analyzes a full 5 mL of solution as opposed to having a 500 μL injection for each sampling period, it has better sensitivity than the PILS-IC and detects some ions that the PILS-IC does not. However, the AIM can for the same reason also saturate the detector at high concentrations, limiting quantification capability while the PILS-IC does not. For example, high ammonia concentrations prevent the AIM from detecting potassium in this study. The AIM for this study used an anion column meant for analyzing carboxylic acids whereas the PILS-IC was set up for the

analysis of common anions. Both instruments were equipped with cation columns for the analysis of amines, but due to very high ammonia concentrations and ineffective performance of the denuder system, long tails for ammonia prevented the AIM from quantifying these compounds. Table 2 shows the mean and ranges for each ion measured by the two instruments during the sampling period.

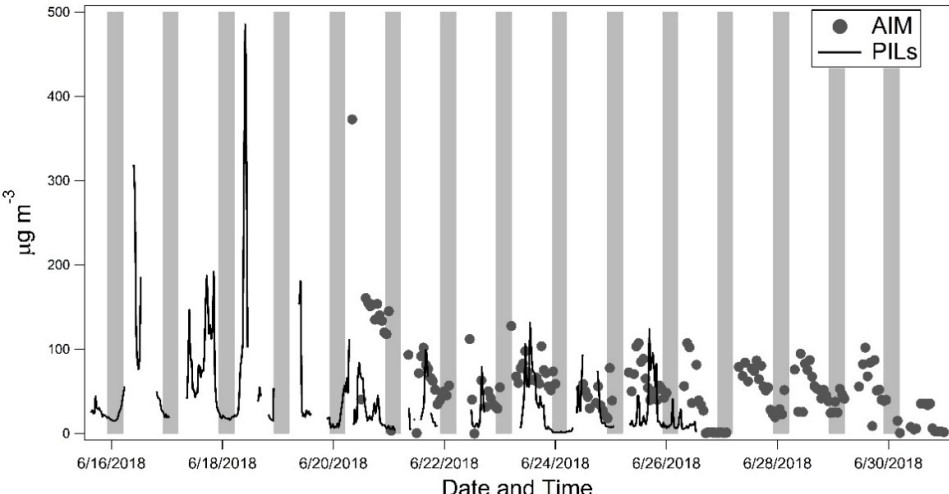

**Figure 5.** Total concentration of water-soluble PM$_{2.5}$ for both the PILs and the AIM during the study. Gray bars indicate lights off (10 p.m.–4 a.m. Central Standard time).

**Table 2.** Water-soluble ions identified during broiler house field study. This is for the time period that the PILS-IC and AIM were running simultaneously, 20–26 June 2018.

| Water-Soluble Ion | Mean Concentration (µg m$^{-3}$) | Standard Deviation (µg m$^{-3}$) | Median (µg m$^{-3}$) | Lowest Measured Concentration (µg m$^{-3}$) | Highest Measured Concentration (µg m$^{-3}$) |
|---|---|---|---|---|---|
| **PILS-IC** | | | | | |
| Acetate | 13.95 | 21.35 | 3.76 | 0.24 | 112.25 |
| Chloride | 0.26 | 0.12 | 0.27 | 0.10 | 0.69 |
| Formate | 1.87 | 1.55 | 1.35 | 0.65 | 7.84 |
| Nitrate | 1.01 | 0.25 | 0.97 | 0.67 | 3.72 |
| Nitrite | 4.07 | 3.86 | 2.40 | 0.84 | 17.10 |
| Sulfate | 1.13 | 2.64 | 0.74 | 0.18 | 36.24 |
| Magnesium | 0.44 | 0.50 | 0.26 | 0.00 | 1.97 |
| Methylaminium | 0.94 | 1.05 | 0.79 | 0.03 | 12.54 |
| Potassium | 1.14 | 1.12 | 0.83 | 0.07 | 4.39 |
| **AIM** | | | | | |
| Acetate | 0.40 | 1.05 | 0.00 | 0.00 | 8.68 |
| Bromide | 0.01 | 0.02 | 0.00 | 0.01 | 0.11 |
| Butyrate | 0.70 | 0.78 | 0.44 | 0.23 | 3.34 |
| Chloride | 0.82 | 2.19 | 0.29 | 0.14 | 15.41 |
| Fluoride | 0.07 | 0.11 | 0.00 | 0.04 | 0.64 |
| Formate | 9.30 | 8.20 | 7.36 | 0.90 | 56.20 |
| Nitrate | 0.90 | 2.47 | 0.31 | 0.12 | 18.77 |
| Nitrite | 0.58 | 0.58 | 0.35 | 0.12 | 2.25 |
| Phosphate | 1.19 | 3.83 | 0.53 | 0.38 | 37.65 |
| Propionate | 9.37 | 10.52 | 5.81 | 1.29 | 57.69 |
| Sulfate | 19.39 | 45.24 | 4.22 | 0.41 | 353.39 |
| Ethylamininium | 1.87 | 5.86 | 0.03 | 0.01 | 36.74 |
| Sodium | 0.31 | 0.46 | 0.02 | 0.00 | 2.24 |

A direct comparison of composition data for chloride acquired from the AIM and PILS-IC instruments is shown in Figure 6. The two instruments are in good agreement for chloride mass concentration through the duration of the study, although chloride is a small fraction of the overall chemical composition detected. The chloride concentration measured by both instruments has a mean of ~ 0.5 $\mu g \, m^{-3}$.

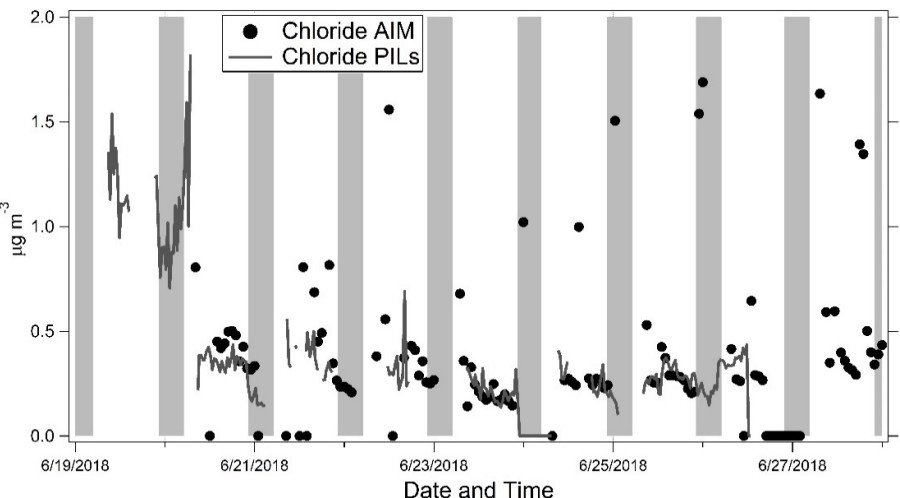

**Figure 6.** Concentration of chloride ions in $PM_{2.5}$ for both the PILs and the AIM during the study. Gray bars indicate lights off (10 p.m.–4 a.m. Central Standard time).

Temporal AIM data for four other ions are shown in Figure 7 as well as the times when poultry house lights turned on and off. Sodium and nitrite ions both show diurnal profiles matching the overall particulate trends, being elevated during daytime hours when lights were on and lower concentrations at night. Other ions on the AIM which show this type of temporal profile include all the carboxylic acid anions (formate, acetate, propionate, butyrate). Fluoride and chloride anions also display this temporal trend. In addition, several cations measured with the PILS-IC that cannot be quantified on the AIM due to the high ammonium signal also exhibit this behavior including ammonium, potassium, and methylaminium.

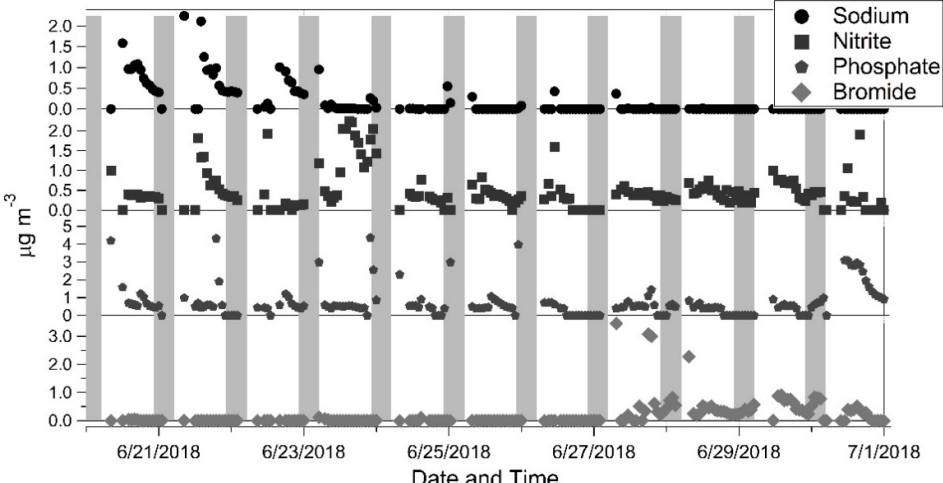

**Figure 7.** Selected water-soluble ion time trends for the AIM during the experiment. Gray bars indicate lights off (10 p.m.–4 a.m. Central Standard time).

Figure 7 also shows data for two other ions for the AIM. Phosphate ion was detectable but low (<1 $\mu g \, m^{-3}$) throughout most of the study. However, on the last day, when the chickens were moved out of the poultry house, the AIM observed a slow rise to several $\mu g \, m^{-3}$.



This was after the PILS-IC instrument was removed from the house, so no data were acquired from this instrument at this time. We believe the high phosphate concentration was an exaggerated effect of animal activity, similar to the daily elevated concentrations observed. The higher phosphate concentrations likely came from the entrainment of deeper dust into the air as the birds were removed rather than daily activity.

Figure 7 also shows the concentration of bromide for the study by the AIM. Bromide was below the detection limit of the AIM method for the entire study up until the last three days prior to bird removal. Starting on 27 June, bromide showed a gradual rise in concentration and then stayed in the detectable range until the end of the study. The reason for the bromide increase is not clear.

Figure 8 shows the time trend of methylaminium and potassium ions from the PILS-IC instrument and sodium and formate ions from the AIM. These cations show the same diurnal trend that the OPC data and several anions showed, rising and then decreasing during the day. The methylaminium data from the PILS-IC are the only particulate chemistry data that are potentially correlated with the new particle formation as it reaches its highest concentrations on that day. During the afternoon of 22 June 2018 (~2:45–4:15 p.m.) a particle spike event occurred as observed by the OPC (Figure A3) data taken during that time.

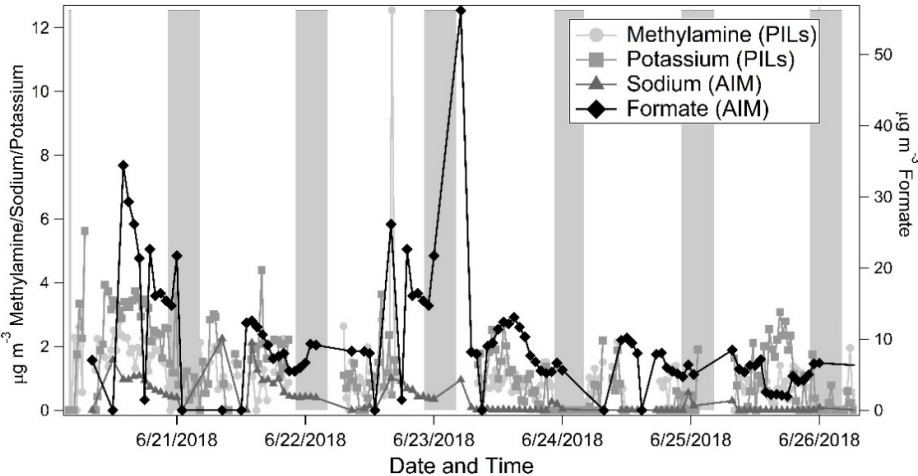

**Figure 8.** Comparison of methylamine, potassium, sodium, and formate throughout the experiment. Gray bars indicate lights off (10 pm−4 am Central Standard time).

As already stated, AIM sodium data are highly correlated with carboxylic acid data, especially formate. The AIM did not measure potassium because of its close retention time with ammonium. However, the PILS-IC instrument did measure potassium. For most days, the time trend of potassium from the PILS-IC is also highly correlated with sodium and formate from the AIM. This is probably not surprising; if sodium is partly present as carboxylate salts, one would expect potassium correlations to be similar.

## 4. Discussion and Conclusions

Emissions studies for particulate matter and its precursors at CAFOs are still rare. In the United States, a National Academy of Science report in 2003 called on the Environmental Protection Agency to perform a systemic measurement study of emissions from CAFOs including for ammonia, hydrogen sulfide, and PM and to develop process-based emission models for the different sectors and waste treatments [37]. As a result of the report, the National Air Emissions Monitoring Study (NAEMS) took place in 2010 and draft process emission models are in development, including for broiler and layer poultry houses [38].

Many observations made during this study are consistent with previous work by other researchers studying poultry facilities (broilers or layers). The water-soluble ionic fraction of particulate matter is only a small fraction of the total mass. This is consistent with other studies where most of the particulate matter in poultry facilities is carbonaceous

in composition and derives from feathers, manure, feed, and litter [20,21]. As expected, in the house, the overwhelming majority of ammonia is present in the gas-phase and not incorporated into secondary PM [12,17,18]. The measured ammonia and particulate matter were generally poorly correlated as has been observed previously [39]. However, a few new particle formation events were observed during the study with one example shown in Figure A2. These corresponded with time periods of overall higher concentrations of ammonia in the house, so this indicates some secondary chemistry occurring inside the house, although it is not the dominant source of particulate matter compared to the entrainment of matter from animal activity.

A number of studies have correlated different chemical compositions in particulate matter in poultry houses with different sources. For example, sodium, potassium, magnesium, and chloride are from animal feed [17,37,38], while nitrate and sulfate are largely believed to be from feces [17,39]. We will not attempt to draw such attributions here, as all we can see is that the daily animal activity significantly increased the PM concentration of most chemical species during the day relative to the nighttime and that animal activity likely entrains all potential sources (feed, feces, feathers, and litter) into the air simultaneously, consistent with a previous study [40]. We could not confirm whether ionic components increased with successive flocks as has been reported previously [41], since we only measured with the full suite of instrumentation for one flock. It has been recently shown that the replacement of traditional wood litter or shavings with artificial turf can significantly reduce entrained PM in the air [42], probably by reducing all components entrained by animal activity. For the last week of flock production, $PM_{10}$ inside the house had a mean concentration of 1360 $\mu g\,m^{-3}$ when lights were on and 342 $\mu g\,m^{-3}$ when lights were off, a factor of four difference. For $PM_1$, the lights on mean was 22.8 $\mu g\,m^{-3}$ and 6.4 $\mu g\,m^{-3}$ when the lights were off.

Although we cannot identify specific sources for each chemical component, there are a couple observations from the PM chemical composition data specifically worth noting. First, several carboxylic acids were identified by the AIM in the PM, including formate, acetate, propionate, and butyrate. Several of these are part of the class of volatile fatty acids (VFA) primarily discussed with respect to their contributions to odor problems from both poultry [43,44], and other production sectors, such as dairy [45]. None of these components were detected by the AIM in the gas-phase during this study, so it is possible that in this case, the odor-causing fraction of VFA is already bound up in the particulate phase and not present in the gas-phase.

Although our intent for this study was not to correlate specific production management with the chemical composition of the PM, as this seems unlikely for most components, the absence of bromide for the entire study until the sudden appearance at the end suggests that there may be some limited situations where the chemical composition of PM can be correlated with specific management practices. The time period of the bromide detection corresponds exactly to when electrolyte was added to the water lines near the end of a flock. An electrolyte is added to keep the birds hydrated just prior to delivery as well as to help clear the water lines near the end of a flock. The electrolyte used for this flock of birds was PWT (Jones–Hamilton) and the major ingredient is sodium bisulfate. Thus, a potential explanation for the increase in bromide is that bromide was cleared from the water delivery system in some form that became volatile and partitioned to the particulate phase due to the abundance of ammonia in the air.

This study was intended to give baseline information on water-soluble components of particulate matter at a poultry house. We found aerosol growth and new particle formation events occurring at the site. Future studies should have a stronger emphasis on the water-insoluble components that make up larger fractions of the particulate mass concentration and elemental/organic analysis which could give more specific information on specific management practices.

**Author Contributions:** Conceptualization, P.J.S., R.M. and K.L.P.-R.; methodology, P.J.S. and K.L.P.-R.; formal analysis, P.J.S., R.D., G.D. and P.L; investigation, P.J.S., T.C., R.D., C.M., G.D., P.L., A.G., D.A.C., C.S., J.W., M.C. and K.L.P.-R.; resources, P.J.S., R.M., M.C. and K.L.P.-R.; data curation, P.J.S., R.M. and K.L.P.-R.; writing—original draft preparation, P.J.S. and K.L.P.-R.; writing—review and editing, P.J.S., R.M. and K.L.P.-R.; visualization, P.J.S., R.D., G.D. and P.L.; supervision, P.J.S., R.M. and K.L.P.-R.; project administration, P.J.S. and K.L.P.-R.; funding acquisition, P.J.S., R.M. and K.L.P.-R. All authors have read and agreed to the published version of the manuscript.

**Funding:** This research was funded under USDA CRIS#5040-12630-006-000D and NSF grant GEO/ATM-1460402.

**Data Availability Statement:** USDA data will be available at the USDA Ag Data Commons Website (https://data.nal.usda.gov/) within 90 days of publication. Restrictions apply to the availability of other data. Data was obtained from the cooperating producer and are available from the authors with the permission of cooperating producer.

**Acknowledgments:** P.J.S. acknowledges the USDA technical support of Anna Foote and Mike Bryant for this study. Mention of trade names or commercial products in this publication is solely for the purpose of providing specific information and does not imply recommendation or endorsement by the U.S. Department of Agriculture. USDA is an equal opportunity provider and employer.

**Conflicts of Interest:** The authors declare no conflict of interest. The funders had no role in the design of the study; in the collection, analyses, or interpretation of data; in the writing of the manuscript; or in the decision to publish the results.

## Appendix A

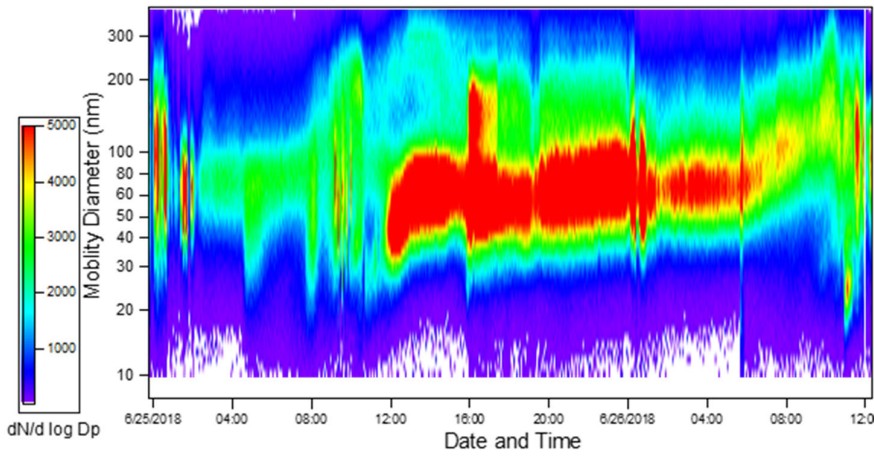

**Figure A1.** Particle number concentrations determined by SMPS from particle growth event on 25 June.

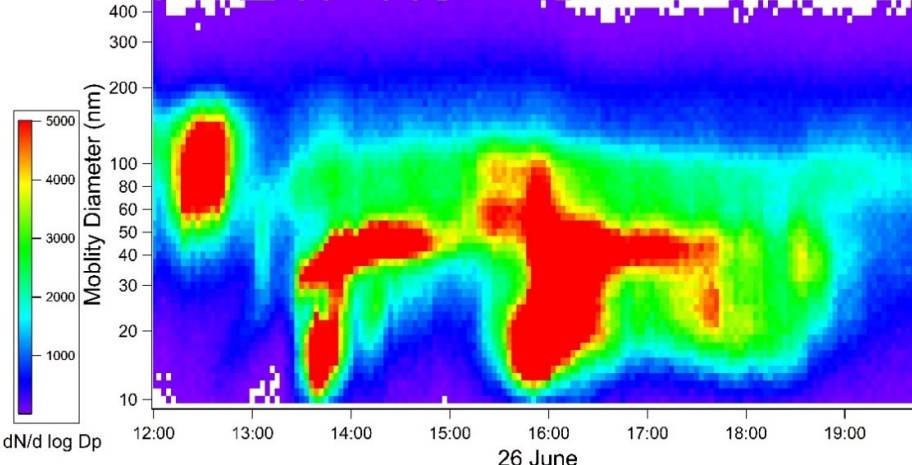

**Figure A2.** Particle number concentrations determined by SMPS from small particle event on 26 June.

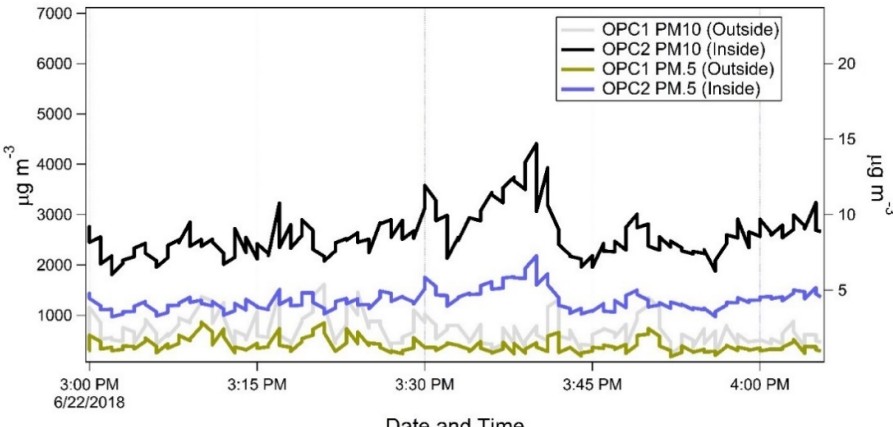

**Figure A3.** OPC data on 22 June 2018 that shows a spike in PM concentrations inside the poultry house in the afternoon between 3–4:15 pm from Figure 8.

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
