# Peer review of "Characterization of Particle Size Distributions and Water-Soluble Ions in Particulate Matter Measured at a Broiler Farm"

_agriculture, doi:10.3390/agriculture13071284_

Round 1
Reviewer 1 Report
Manuscript Number; agriculture-2366581
Title; Characterization of particulate size distributions and water-sol- uble ions measured at a broiler farm
Evaluation; Minor Revision.
1. Figure 1 needs to revise.
2. Materials and Methods
“Ions calibrated for during this study were acetate, chloride, formate, nitrate, nitrite, sulfate, magnesium, methyla-minium, and potassium”.
Why author study these ions? Any standard protocols?
Add more about the QA/QC e.g. MDLs, repeatability etc.
3. In all of the main text, many numeric data are given with too many significant figures; 2 significant figures suffice, and 3 suffice in case the first significant figure is "1". e.g. Table 2, many digit data to use.
4. Overall, the method is not clear, and it does not provide clear results, so it is judged that it should be re-examined after rewriting. Including the above information, please reconfirm and revise the paper.
5. Conclusion; Many paragraphs are too short. Please revise and combine them into only one paragraph in the conclusion. The conclusions could be further developed, there is a lot of interesting data in the article.
Moderate editing of English language need.
Reviewer 2 Report
Dear,
This scientific article addresses the characterization of particles present in the air environment of broiler facilities. For this type of research, it is expected to have some innovation in terms of methodologies and methods that enable the monitoring of emissions in the facilities in a simpler and more accessible way. I believe that characterizing using specific methodologies for the academic field, which are usually complicated and costly, does not contribute significantly to the emissions monitoring process. I would have liked to find in the manuscript a description of the relevance and originality of this work since it is not clear what the authors bring as new. Additionally, I did not have access to the cover letter of the article, which should provide an explanation of why the content of the article is significant and what the findings are in the context of existing work.
The lack of line numbering in the text, as shown in the model provided by Agriculture, considerably hinders the reviewer's work, as it is impossible to make specific comments and later verify whether changes have been made or not. Therefore, I conducted a general evaluation of the manuscript.
The title needs to be improved to be more specific, relevant, and accurately represent the research topic.
The abstract does not provide the research context or the significance of the work. It should include at least an introductory sentence highlighting the relevance and originality before presenting the objectives. The information presented in the abstract is superficial.
There are terms in the Keywords section that are already part of the title.
In the Introduction section, there is a focus on "animal feeding operations," but this is not the only or main cause or source of atmospheric emissions in animal production facilities. For example, ammonia is more related to bedding decomposition and waste presence than feeding operations. There are misconceptions that convey a contradictory idea to the reader, such as "More recent studies show that emissions from agricultural facilities can cause nitrogen enrichment of soils and waterways." There are already protocols and emissions inventories adopted by various countries, which contradicts the information presented in the second paragraph, "Traditionally, agricultural sources of emissions have not been included in air quality regulations because of a lack of data and the difficulty of mitigation." Justifications are lacking to defend the relevance of this work. The mere need to understand the processes does not justify conducting a scientific study. The authors need to conduct a better literature review because there are numerous studies conducted over a decade ago on the characterization of emissions in broiler chicken facilities. An example is the citation used (22) in "Not as much research has been done to understand emissions from meat chicken sheds, as opposed to egg laying houses, but odor emissions are thought to be dependent on chicken litter conditions, such as moisture and microbial content of the litter," which was published almost 10 years ago.
More information and details are needed about the characteristics of the facilities. The figures used in the Materials and Methods section have poor quality and do not add relevant information to the text. There is insufficient information about the equipment used to replicate the research. The characteristics of the sensors used are not provided. The stated objective does not justify the need for multiple equipment with the same function for characterization. No experimental or statistical design is presented. There are conclusions about methodologies that are repetitive, contradictory, and lack scientific basis.
A concise and precise description of the experimental results is missing. The presented results contain flaws that do not correspond to the reality of the facilities and contradict what already exists in the literature. Scientific support is lacking to substantiate the information presented in the results. There is a significant amount of literature review in the results section without citations.
There are discussions that are not related to the objectives. There is no clear conclusion that addresses the objectives.
There are many references that can be considered outdated throughout the manuscript, and the literature search conducted by the authors was inefficient since there are already many works that have made significant advancements in this line of research, and they were not even included in this manuscript.
Reviewer 3 Report
- why this experiment only took place between the 15th and 30th of June and not the whole period of chickens being there (14th of May to 30th of June)?
- table 2 - please rewrite numbers using only significant digits
Round 2
Reviewer 2 Report
Despite the changes made, I am still not convinced of the relevance and originality of the work. There are still flaws in some sections of the manuscript, mainly in Material and Methods, which compromise its quality.